# BL0K: A New Stage of Privacy-Preserving Scope for Location-Based Services

**DOI:** 10.3390/s19030696

**Published:** 2019-02-08

**Authors:** Abdullah Albelaihy, Vijey Thayananthan

**Affiliations:** Department of Computer Science, King Abdulaziz University, Jeddah 21589, Saudi Arabia; vthayanathan@kau.edu.sa

**Keywords:** privacy, location-based services, bloom filter, zero-knowledge proof, BL0K

## Abstract

Location-based services present an inherent challenge of finding the delicate balance between efficiency when answering queries and maintaining user privacy. Inevitable security issues arise as the server needs to be informed of the query location to provide accurate responses. Despite the many advancements in localization security in wireless sensor networks, servers can still be infected with malicious software. It is now possible to ensure queries do not generate any fake responses that may appear real to users. When a fake response is used, there are mechanisms that can be employed so that the user can identify the authenticity of the query. For this reason, this paper proposes Bloom Filter 0 Knowledge (BL0K), which is novel phase privacy method that preserves the framework for location-based service (LBS) and combines a Bloom filter and the Zero knowledge protocol. The usefulness of these methods has been shown for securing private user information. Analysis of the results demonstrated that BL0K performance is decidedly better when compared to the referenced approaches using the privacy entropy metric.

## 1. Introduction

The proliferation of devices, especially smartphones, has popularized the use of location-based services (LBS). The use of smartphones has grown significantly recently and plays a prominent role in the growth and popularity of LBS [1]. As a result of the development of new communication means, the information gathered from LBS has given users a partial advantageous consciousness of their environment [2]. Numerous Android or iOS applications allow the user to download then use LBS to upload queries to the location-based services servers [3].

The users are able to receive LBS information by using points of interests (POIs), from which they can also look for restaurants or stores within their geographical area and confirm product prices. This feature has popularized LBS, as it provides numerous users simple solutions for both location sharing and awareness, making LBS resourceful and advantageous [4].

Although LBS has numerous advantages, it presents some inherent challenges in finding a balance between two requirements that are somewhat diametrically opposed: efficient response of the server and security and user privacy when answering the queries of the users. Users tend to ask for correct answers to the queries that are location-based, while the users expect their information to remain secure and private [5]. The dilemma is that when no location is mentioned, the server cannot generate a response to the location-based query. For instance, a user cannot expect a response to the query “What are the closest gas stations near address X”, if they are not willing to provide their specific location. This challenge revolves around achieving the delicate balance between acceptable user privacy and a fast response to queries. LBS queries that are submitted by users have to ensure privacy and be efficient. Unfortunately, these queries can often lead to issues with security, which were the first issue with implementations: whether data-stores (services, databases, and websites) can be queried in order to display a minimal amount of information to the data-store while providing maximum utility (availability), to the user making the query.

If you assume that the server may be compromised or even malicious, a second issue arises: the possibility of making a query with an unforgeable response, which means that the server is unable to generate a fake response that seems to be a legitimate response. If the server generates a fake response, the recipient of the response may recognize that it is fake (integrity).

The third issue is that any information leak in the security system is not good. If each query leaks a bit and thousands of queries are made, the attacker can learn some information that must remain confidential (confidentiality). The bloom filter, which is discussed in this paper, is an example [6,7,8,9].

These services obviously pose a serious risk to user privacy by potentially disclosing the user’s location to the service providers, and the location information is also disclosed over queries that are subsequently provided for the purpose of gathering LBS. Privacy protection of users with respect to LBS providers is crucial when making sure that the LBS ecosystem is both effective and secure. This will lead the LBS market to continue to flourish and develop, as well as the users may soon find themselves to be more comfortable with the use of LBS. In addition to its series of benefits, LBS still pose serious risks to user privacy. By gathering information concerning a user’s location, sensitive privacy information could be deduced that is related to the recipients of the LBS service. There are two kinds of privacy LBS-related issues: location and query privacy. For example, a user that lives in a rural location might disclose their exact location information within the size of the rural area, which will prevent preservation of the user’s location privacy [10,11,12,13]. Consequently, malicious actors (and in particular, eavesdroppers) can easily form correlations over a number of queries that can inadvertently lead to an unacceptable loss of privacy. Therefore, a solution for the preservation of privacy must also consider the potential for malicious behavior [14,15].

It is also important to consider location-based information. A reply to an LBS-based query is not the answer to one information item but a group of information items. Due to this functionality, eavesdroppers and other malicious actors can easily create relationships between different queries, which can later lead to a loss of privacy. Any particular solution that tends to preserve privacy should consider malicious acts. Many techniques have been suggested as optimal solutions for maintaining confidentiality in the LBS [16,17].

There are numerous possible methods of creating solutions to issues that are based on privacy. The primary aim is to ensure the efficiency and confidentiality of LBS and their queries. These requirements tend to raise some questions about server security, mostly LBS. One alarming issue is that the servers can be penetrated, so the servers are under full control by the attacker, resulting in mischievous activities, including generating queries to responses and respond to queries. In case that server can generate a fake response to recipient, so the fake response can be identified by the recipient [18,19,20].

Shen et al. [21] conducted research on mobile device online social networks (mOSNs), which offer a local distribution service. Researchers studied the present issue in location sharing and later suggested that BMobishare, which combines a dummy location technique and bloom filters (BF) between LBS and social network server (SNS) within the Wi-Fi tower. BMobishare has an improved safety mechanism that can ensure location discretion. The authors used the BF to increase the security of sensitive information. Though this tactic safeguards confidential information, it is faulty. There is a risk of leaked information amid the servers and users, which can increase the overhead and decrease entropy.

Albelaihy et al. [22] proposed an approach that combines the BF and oblivious transfer (OT), thus creating the BLOT technique. In this situation, the sender is the LBS, whereas the user is the recipient when the user starts the procedure by transferring a group of q-queries to LBS. Hence, the queries create a friendly a to produce q-answers, e.g., the message m0…m (q–1).

Therefore, a benign LBS cannot estimate the q-queries that are relevant, apart from a prospect of 1/q, proving that BLOT improved the current security in communication amid the user, the server, and while averting attacks. This approach protects against eavesdropper only, but the drawback of using this method is not being able to detect the malicious servers directly.

To overcome the issues described above, we propose the BLoom Filter 0 Knowledge (BL0K) combination method, which consists of bloom filters with zero knowledge. By implementing BL0K in the LBS scenario, the user plays the role of the verifier, while the LBS server plays the role of the prover. The zero-knowledge proof (ZKP) in cryptography can secure the communications and hide important information. ZKP consists of two phases: initialization and challenge/response.

The initialization phase is an exchange that takes place when the user first contacts the LBS. The challenge/response phase is executed after the successful initialization. This phase can be executed multiple times, so in any situation where the response fails to match the challenge must be the result of a malicious server. The worst action by a malicious LBS under a ZKP configuration is providing a denial of service, thus securing authentication between user and server against attacks (malicious LBS). Therefore, BL0K is expected to enhance the communication security between user and server, protecting the server against attacks. So, the contributions of BL0K can be summarized as follows:
(1)We propose a framework called BL0K that creates an entirely new encryption communication procedure in the problem domains (malicious server) and a secure application for use in resource-constrained schemes such as mobile devices.(2)BL0K is capable of reducing the amount of exposed data during an attack, hence BL0K can be used to lessen data leakage.(3)BL0K can be used to make the outputs of the algorithm appear completely randomized. Thus, increasing the general performance level that would also increase the level of privacy.(4)We evaluated the performance of BL0K framework: using a dataset close to the real-world queries by comparing its performance with the BLOT in terms of privacy (entropy), and the reported performance improvements increased to 93% (averaging 25% to 30%).

The rest of the current paper provides the scheme to overcoming the issues described earlier using ZKP and BF. Related work is reviewed in Section 2, Section 3 provides the preliminaries with all its subsets. The BL0K technique is outlined in Section 4. BL0K performance measurements, calculation of the performance entropy for BL0K, the BL0K performance results, and conclusion are provided in Section 5, Section 6, Section 7, and Section 8, respectively.

## 2. Related Work

### 2.1. Privacy of Bloom Filter

The role of social networking services tends to depend on the flexibility and availability of sharing locations with respect to privacy protection. Cox et al. [23] developed opaque identities that were designed to help with the capability of sharing a person’s presence amid trustworthy friends and untrustworthy strangers. Though the prior work resolved the problem of the lack of flexibility in the sharing of the locations regarding privacy protection, the stated method’s strictness stops when it is directly used [24].

Likewise, a mechanism referred to as Mobishare [25], proposed by Wei et al., has a privacy-preserving ability that is extremely flexible in sharing locations with both untrusted strangers and trusted social relations in mobile online social networks (mOSNs). Nevertheless, a Mobishare method is actually an extension of Smoke Screen though it uses an imitation tactic that included a fake identity, prevented revealing the user complete locations and identity amid social network providers and LBS suppliers. A later enhancement by Liu et al. produced the ability to spread encryption designed to stop users’ location discretion by letting users remove and add friends [26].

Although these improvements were implemented to address the numerous assaults on the LBS provider, an attacker can still find a way to access the information about the friends’ relation to the user and their true time location. The Paillier-Cryptosystem, established by Li et al., might have stopped the problem. Li et al. introduced the idea of using Mobishare+ mechanism to employ a remote set of connection protocols to stop the social network server from learning individual data. Though this mechanism improved security, Mobishare+ needs to execute many complex coding and decoding operations that increase the overhead computations [27]. Other approaches are encryption-based, mainly in the cloud, that are used for the defense of privacy [28,29,30].

Palmieri et al. [31] disclosed how the availability of cheap positioning systems has made it possible for them to be embedded in devices and other smartphones. This has led to the development of location-awareness applications, allowing users to demand personalized services that are dependent on their geographical location. However, only a small amount of information concerning the user’s whereabouts should be available at any given time, since a user’s geographical position is highly sensitive. Several applications such as the navigation system are dependent on the user’s movement and require constant pursuing. Other applications only require knowing a user’s position in relation to an individual area of interest.

In Ahmadi et al. [32], BF are simple and space-efficient randomized data structures used to represent a certain group, such that any connection requests are maintained. Recently, the acceptance ability in their database and networking applications has been improved.

A BF is composed of two major phases: programming and the membership request. From there, a new method for reducing hash table (HT) access time is introduced, which further integrates the HT into the filter. Hence, after the BF used for the external entry of a program, the following item is stored in an HT simultaneously. This method has the advantage of minimizing the average bucket size, maximum search length, and impact quantity.

Calderoni et al. [33] addressed issues related to the security of user location data through the use of BF, which can be described as a compressed data structure that is usually used to represent numerous groups. So, the spatial bloom filter (SBF) tends to be an expansion of the standard BF. The SBF was built with the intention of improving protection of user privacy through the management of the spatial and geographical information. Additionally, the finished creation consists of multi-party protocols that have the capability to preserve the privacy of data based on the locality, which are identified by the public encryption arrays. This procedure is often viewed as an efficient method of protecting the privacy of users when looking at their exact location. Parts ∆_1_, ∆_2_, and ∆_3_ are used to build an SBF, as shown in Figure 1, where the hash functions 1, 2, and 3 are used to link all elements to the filter, except for the first 10 elements of the SBF [33].

In Dong et al. [34] the protocol named Private Set Intersection (PSI) was introduced based on the intersection of OT and BF. A PSI is linearly complex and, for the most part, relies on the efficient symmetric key operations. The authors stated that their method provides high scalability due to its basic protocol capability as well as an enhanced protocol, which was proven in the semi-honest and the malicious model, respectively. BF has been proposed as a probabilistic data structure used for hiding sensitive data. Unfortunately, each request leaks a maximum of one bit of information and the hash function requires careful design and security. BF analysis is focused on the orthogonality and independence of the structure. The results show that the proposed hash functions are less dependent and permeable than the comparison method, while significantly improving performance [35].

### 2.2. Privacy of Zero Knowledge and Localization

Zhu et al. [36] focused on LBS in a vehicular ad hoc network (VANET), where they explained the relationship between anonymity degree and expected company area, along with vehicle density. The authors then developed a privacy mechanism based on the multi-party computation and provided the cloaking region (CR) between vehicles. The scheme involves computing each vehicle based on the CR and sending it to the LBS and then returning a point of interest. The authors also added ZKP to detect the malicious vehicle.

Another use of ZKP was proposed by Xiao et al. [37], where a novel zero-knowledge multi-copy routing algorithm with homing spread (HS) was used in mobile social networks (MSNs). The authors focused on community homes spread messages for optimally distributing a number of messages.

A similar concern was addressed by Zhu et al. [38], where they highlighted the adversary guesses of passwords along with proposing the enhanced Kerberos protocols based on public key cryptography. However, this solution was found to require a significant amount of computation resources. As such, a new enhanced Kerberos protocol and non-interactive ZKP was introduced in the authentication process without revealing any information.

Jagwani et al. [39] applied the middleware to help generate the certificate for a client in the form of a pseudonym, which is used by a Location-Based Service Provider (LSP) at the time of seeking a service to solve the problem in the middleware under bottleneck. ZKP was adopted for authentication. Kotzanikolaou et al. [40] proposed an anonymous authentication system. This method assigns network operator roles as an anonymous accreditation issuer for their users. However, this system supports the accreditation and is not flexible for transfer without certain security settings. This system uses the standard primitives of ZKP, message authentication codes (MAC), and Challenge-Response. Mainanwal et al. [41] suggested the combination of ZKP and RSA cryptography algorithm called Z-RSA used for the authentication in the web browser login system to prevent attacks on user passwords.

Privacy-preserving data exchange algorithms are active research areas. Although recent breakthroughs such as fully homomorphic encryption (FHE) promise extremely strong privacy guarantees, such approaches are extremely inefficient at this time. Some location protection tactics, such as ZKP, are used for improving the strength or even protocols for maintaining safety regardless of the compromises. ZKP lets the participants prove any messages that are sent, which are properly produced within the procedure, are without any form of disclosure of any secret data [42]. ZKP proofs are often viewed as zk (M; N). The S in the ZKP is a section of the Boolean formula built in by using cryptography and is shown as zk: S (N_1_ and N_N_; M_1_ and M_M_) and is even a portion of the two distinct strings of terms, whereas the holders of the location a_i_ and b_i_ show the relations of M_i_ and N_i_, correspondingly. The module in M_i_ cannot be exposed after the instantiation of the placeholder “S (M/a) (N/b)”, which is often revealed to be true. After this, zero-knowledge profits can then be regarded as successful. For instance, zero knowledge zk_check_ (a_1_; b_1_) _a_2_ (sign (m;k); m; vk(k)) tends to prove that the information of a signature can be successfully verified with a key vk(k) [43]. ZKP tends to vary from the traditional cryptography, where they tend to authenticate and secure the privacy of communication in contemporary applications such as LBS. The numerous benefits of ZKP often supercede the embryonic techniques of security and confidentiality since they can confirm a particular user and still ensure that the user is unidentified [44]. Although ZKP is mainly desirable for defending the confidentiality of a user and the content of a message, computer-assisted support in modeling the security protocols is not available. However, for these intentions, this protocol is still vulnerable [45]. Garg et al. [46] proposed a new category for localization method in wireless sensor networks using localization algorithms that contained based on multiple key features such as Anchor Based, Anchor Less, Range Based, and Range-Free.

Khadim et al. [47] picked two methods to compare performance, the first of which was Grid Location Service (GLS) and the second was Hierarchical Location-Service (HLS) beside to the Greedy Perimeter Stateless Routing (GPSR) routing protocol. Network simulator (NS2) was used for wireless sensor networks to test the performance and the request travel time (RTT) as well as the query Success ratio (QSR). Another study [48] reviewed the many advantages of LBS that affect the usage of intentions that then affect users and innovativeness for LBS. So, this paper was developed using the concept of technology acceptance behavior, which was then applied to LBS users to specify LBS usage intention, as well as technology acceptance behavior opinions. Hasan et al. [49] preset the problem of personal privacy in LBS, proposed a privacy architecture, and focused the bounded perturbation technique on keeping the trajectory of each user from the privacy gaps.

## 3. Preliminaries

In this section, we present the deficiency in the BLOT approach, zero-knowledge, and bloom filter in LBS. Before we explore BL0K in detail, we provide an overview, research design, and the architecture of BL0K.

### 3.1. Deficiency of BLOT

The use of ZKP allows data to be transmitted cryptically between the user and the LBS server. The BLOT approach protects against eavesdroppers but does not provide perfect protection. The deficiency of this approach is not being able to directly detect the malicious servers. Hence, continuous ambiguity must be maintained, which may affect the performance and data leakage.

### 3.2. Zero-Knowledge Plan within LBS

A data-transfer protocol between LBS user and LBS server must obey the following properties:
(1)Proof of an assertion: Provided by the protocol regarding LBS information with probability p.(2)Probability: By increasing the queries, probability q can be increased exponentially close to 1.0.(3)Leakage: No information is leaked by any query, except with negligible probability.(4)Malicious LBS server: Using probability q, it is possible to detect a malicious LBS server.

ZKPs are a cryptographic technique that enables a prover (P) to demonstrate to a verifier (V) that P possesses k bits of information (k arbitrary), without exposing even a single bit of this information to V. As shown in Figure 2, the initial development of ZKP relies on an interactive challenge/response protocol that could be used to demonstrate the knowledge of a single bit of information [50,51,52]. The ZKP technique has since been extended to completely automated (non-interactive) protocols that can handle an arbitrary bit vector of length k that is known only to the prover P. ZKPs are probabilistic protocols. In a typical implementation of a single ZKP, iteration can be used to provide 50% certainty that P knows 1 bit of information [53]. For N iterations over a k-bit vector, the probability provided by the proof of knowledge scales as (1 – 2^−N^)^k^. Traditionally, the verifier V is referred to as Victor, and the prover P is referred to as Peggy. In the LBS case, the verifier is the user (receiver), and the prover is the LBS provider (sender).

Note that while the prover P must be honest in the first case, it is not necessary for the verifier V to be honest. V can conduct a denial-of-service attack against the prover P by providing inaccurate reports on the outcome of a particular ZKP iteration, but P cannot determine any component of the k-bit vector K possessed by P with better chances. For a second case, even if a malicious prover P offers a putative vector K’, a single iteration of the ZKP algorithm can be used to demonstrate that K’<i> == K<i> with random probability at best (here <i> denotes the *i*th bit of the vector). Thus, the malicious prover is incapable of forging information possessed by the prover P and is also incapable of forging proof of knowledge of that information. After N iterations of the ZKP algorithm, the malicious prover has only a probability of 2-Nk of providing knowledge or proof of knowledge for a k-bit vector known as the prover P [54].

The ZKP approach has immediate relevance to the LBS privacy problem. In the LBS scenario, the user plays the role of verifier V, while the LBS server plays the role of the prover P. The user does not only submit a query Q; instead, the user submits a query Q together with an assertion A. A benign LBS will provide either a proof or refutation that the tuple (Q, A) is consistent with some probability. The user may submit additional query-assertion pairs in order to improve the likelihood that the response is accurate to the degree that satisfies the user’s expectations. A malicious LBS will only be able to provide a self-consistent proof or refutation with negligible probability. The only action that a malicious LBS can take is to provide a denial of service, e.g., by dropping queries or providing nonsense responses. The likelihood that the user can detect a malicious LBS increases by at least a factor of 2 for each (Q, A) tuple [55,56,57].

### 3.3. Bloom Filter Plan within LBS

The user, LBS server, and SNS should have the following properties for their protocols:
(1)User/server: The user and SNS can send a particular query if there is something in the LBS server’s range.(2)Query list: The original server can locate one bit of data from the query list, while SNS can locate none.(3)Server range: Set of objects in the range of the server will need to be itemized.(4)Malicious LBS server: A malicious LBS server can be able to detect with probability q.(5)Probability: Probability q can be increased close to 1 by increasing the queries.

The BF facilitates the testing of a secure membership and never discloses more than a bit of data concerning this set. As depicted in Figure 3, if A is an element that presumes a set S, then the BF generates a probabilistic system that can be used in determining if “q is not a component of S” and H(q) = 0, which means no false negatives occurred. Thus, the assertion is accurate. When all k values in the hash functions are 1, but the item q is not in S, false positives occur. B denotes bloom filter; H denotes hash function [58].

LBS embraces BF, which can use data about a location set as bit vectors. A user most often creates q pairs (assertion, query), which are then transmitted to the LBS, which then creates a vector of q qualities. If the *i*th points of this established vector are zero, then the resulting Qi and Ai are unpredictable. However, if the *i*th point is 1, it follows that Qi and Ai are steady [33].

The assessments can be correct even if the server is benign. Conversely, a malicious server results in erroneous assessments. As previously established, the client can build Q and A pairs with an identified state. It follows that the server can run tests against recognized values. False positives, which most likely point to an erroneous server, can be decreased using N reiterations of q series. Therefore, a server molded to have a malicious event can be identified by the property (1 – p^N^), where p refers to a genuine false positive. Apart from a negligible probability caused by the inverse HK, malicious servers cannot receive information from any exchange.

### 3.4. Overview of BL0K in LBS and Research Design

In the proposed approach, we investigated two protocols to protect privacy: ZKP and BF. The aim of this approach is to decrease the data leakage and preserve the user’s privacy queries in LBS. So, the combination of BF and ZKP can be fully used using a two-stage method as shown in Figure 4.

Therefore, the first stage s implemented with ZKP, which considers the case where there are two parties, V and P. V has some plaintext and P has access to an encrypted version of this plaintext. So, P wants to prove that it can decrypt its encrypted version using a zero-knowledge method. Clearly, it wants to prove that it can decrypt the encrypted version but without any secret data being shared between V and P. The second stage executes the BF to mask the sensitive data between SNS and LBS.

### 3.5. Security for LBS: The BL0K Protocol

First, the component protocols are depicted. The BL0K method is in the form of a framework. Within ZKP + BF, the ZKP in the LBS scenario grants the user the role of the verifier V, while the LBS server plays the role of the prover P. The ZKP has two phases of execution: an initialization phase and a challenge/response phase. This method makes use of a BF, wherein LBS has a BF encoded with location information by bit vectors, BF includes multiple hash functions, and the hash functions show the bits of a member x within bits that are set in the BF to create a security mechanism between the LBS and SNS, protecting each other’s privacy. The user produces q pairs (query, assertion) and ZKP creates a security technique between the user and LBS prover. This ensures privacy without leakage. The queries are then transmitted to LBS, resulting in bit vectors of q values. The ith position of the vector is 0, Qi and Ai form an inconsistent pair. If the ith position, however, is 1, then the pair is probabilistically consistent. Thus, if the LBS server is trusted, each subsequent evaluation will be accurate within the limits of the BF algorithm. It follows that only some evaluations will be accurate when the LBS server is malicious.

### 3.6. BL0K System Architecture

The LBS scenario shown in Figure 5 has four objects in the mobile devices online network.

First, user A can gain entry with internet speed of either 3G, 4G, or 5G now, later share their whereabouts, and then ask about the friend’s position. Secondly, SNS is able to manage the identity-related data of the user that include a backing store, friend lists, user profiles, etc. Thirdly, the LBS often stores the anonymized data of the location of the user and gives the LBS, per the request of the user, the real-time locations of nearby people. Lastly, the CT is oriented to aid the communication of the user with LBS and SNS. BL0K assumes that LBS, CT, and SNS are linked with the links that are secure and high speed.

### 3.7. ZKP Standard

Consider the case where there are two parties, V and P. V possesses some plaintext and P has access to an encrypted version of this plaintext. P wishes to prove that it can decrypt its encrypted version using a zero-knowledge approach. Specifically, it wishes to prove that it can decrypt the encrypted version without leaking any secret information between V and P, in either direction.

ZKP must have three characteristics: completeness, soundness, and zero knowledge. These are formally defined as follows [38,59]:

For completeness, if the notification is genuine, the honest user (verifier) will be persuaded by the honest LBS (prover) using the following equation:(1)$$Y_{Z}^{2}\underset{e_{j}=1}{\prod }V_{j}\;\left(mod\;n\right)={\left(r\underset{e_{j}=1}{\prod }S_{j}\;\right)}^{2}\times V_{j\;}\left(mod\;n\right)$$
(2)$$=r\underset{e_{j}=1}{\prod }(S_{j}^{2\;}\times V_{j\;})\;\left(mod\;n\right)$$
(3)$$=r^{2}\;mod\;n$$
(4)$$=X_{z}$$

Soundness: If the notification is false, the dishonest prover cannot convince the honest verifier that it is true, except with negligible probability.

For Zero-Knowledge ZKP, if the notification is true, no cheating verifier learns anything other than the statement is true. In this case, Peggy is making the statement “I can decrypt, or I have access to a device or third party that can decrypt.” We add a fourth requirement to the system: that it must be independent of the type of encryption/decryption being used. In particular, it must work for both symmetric and asymmetric crypto, without any change in the ZKP.

In this paper, we refer to the plaintext held by Victor as s. We use s to emphasize the fact that the plaintext must be protected. Not even one single bit of s can be transmitted to Peggy. The corresponding encrypted text that Peggy possesses or has access to we refer to as c. Her key material we refer to as K. Part of K may be publically known (e.g., the public part of an asymmetric key), but in general we assume that K must be protected so that not even a single bit of K can be transmitted to Victor.

We make one assumption regarding the cryptographic intractability of certain operations. The assumption is that the “sqrt mod n” problem is cryptographically intractable. Let p and q be primes such that p ≡ 3 mod 4 and q ≡ 3 mod 4. In this case, it is not acceptable for either p or q to be approximately prime (this term is defined below).

We compute n = pq. The problem of the “sqrt mod n” determines, for a given y and n, whether there is an x such that x^2^ ≡ y mod n. The problem is constructive: if x exists, then the algorithm must produce it. There is no known polynomial time algorithm that can solve this problem. In this formulation, the key material is p and q and the plaintext is x. It can be proven that the intractability of this problem is equivalent to the intractability of factorization.

Given plaintext s, we require an invertible function g such that g(s) is a unique integer at least 256 bits in length. Function g must be polynomial time computable, the inverse g^−1^ must also be polynomial time computable, and it must comply with the following: g^−1^(g(x)) = x and g(g^−1^(y)) = y for all plaintexts x and for all encoded integers y. This requirement is easy to meet. Length-encoded values with padding, as defined by RFC1321 for example, can be used. It is not intended that g or its inverse contain any key material. It is an advantage for both V and P that the complete algorithms for g and g^−1^ are made publicly available. Standard encoding methods consider time as linear. For the rest of this document, we refer to s and g(s) interchangeably.

There are many different forms of the ZKP algorithm, ranging from the very simple Ali Baba protocol [50] as well as many more complicated algorithms with specific properties. We use a version of the Gaillou-Quisquater (GQ) zero knowledge algorithm [60] with an auxiliary secret. During the initialization phase, V is provided with s and P is provided with c (presumably from its database).
(1)PEGGY STEP: Peggy generates primes p and q, such that p ≡ 3 mod 4 and q ≡ 3 mod 4. She computes n = pq and sends n to Victor. The values p and q are the “auxiliary secret” referred to above.(2)VICTOR STEP: Victor chooses a random integer x and computes w = (xs)^4^ mod n. He then sends w to Peggy. Note that because of the intractability assumption, it is not possible to recover a single bit of (xs)^2^ or (xs) from w. Good practice suggests that Victor uses a new value of x for each ZKP interaction (LBX transaction).(3)PEGGY STEP: Peggy can now compute the principal square root y_1_ of w mod p and the principal square root y_2_ of w mod q. Using Fermat’s Little Theorem [61], these values are:
(5)$$y_{1}=w^{\left(p+1\right)/2}$$
(6)$$y_{2}=\;w^{\left(q+1\right)/2}$$Peggy then uses the Chinese Remainder Theorem [62] to construct a value y such that y ≡ y_1_ mod p and y ≡ y_2_ mod q. It follows that (xs)^2^ ≡ y mod n. Peggy now needs to find m—the multiplicative inverse of s^2^ mod n. Euclid’s algorithm tells us that m exists and is computable in linear time if and only if gcd(s^2^, n) = 1. This will be true, except with negligible probability. Peggy then computes m and observes that m(xs)^2^ ≡ x^2^ ≡ *my* mod n. Peggy sends *my* to Victor.(4)VICTOR STEP: Victor computes x^2^ mod n and observes that this is equal to the value just received from Peggy, namely *my*. First, consider that this is a probabilistic algorithm. The likelihood of success is 1 − 2^−b^ where b = log2(n). This may increase arbitrarily using larger values of p and q. Secondly, in Step 3, Peggy did not need s^2^; she only needed m, the multiplicative inverse of s^2^ mod n. By the first assumption, it is computationally intractable to recover any part of s from m. (his is why a fourth power was used rather than a second power. The value m can be safely be stored in a database referring to the quantity c, or some abstract token that refers to c. If the encryption was performed on a device, then it could be arranged that the encryption mechanism returns m and a token that points to c. Finally, we must discuss the issue of Fermat Liars and “approximately prime” values. Fermat’s Little Theorem states that if p is prime, then a^p−1^ ≡ 1 mod p for all such that 1 ≤ a < p. If n is composite, then a^n−1^ ≡ 1 mod n, where n is referred to as a Fermat Liar or an approximately prime integer. For example, n = 221 is a Fermat Liar for a = 38. This is why the auxiliary secrets p and q must be prime, and not approximately prime. Any primary algorithm that runs on p and q must ensure that they are truly prime. Since p and q are computed offline, this does not affect the runtime performance of the system.

Note that the ZKP standard describes an abstract transfer protocol for learning information about a quantity (an LBS data item) without actually transferring the information itself. Like the OT protocol, cryptographic primitives are combined to yield a transfer protocol that is immune to information leakage, except with negligible probability. The system architecture is outlined in Table 1.

The notations are outlined in Table 2.

## 4. BL0K Technique

In this section, we detail the BL0K method, which sufficiently improves the privacy of the system through gathering BF with ZKP.

### 4.1. BL0K Querying Users’ Locations

Figure 6 shows the flow of the BL0K approach. ZKP constructs a security technique through the users and LBS prover to ensure that privacy is guaranteed and to avoid the leakages. BF creates a security technique over LBS-SNS, and both are protecting each other’s privacy.

### 4.2. BL0K Algorithm

#### Zero Knowledge with Bloom Filter Scenario (BL0K)

Step 1: Initialization of ZKP as shown in Algorithm 1:

Prover has an ID^A^ that can calculate J(A). It also has a secret ID. Before step 2, we have two files to update: one for BF updating and this function updates the BF with the data in a text file “DataListforBF.txt,” and one for SNS backing store with the data and its locations as (SHA32 bit) code.
**Algorithm 1:** BL0K Algorithm.
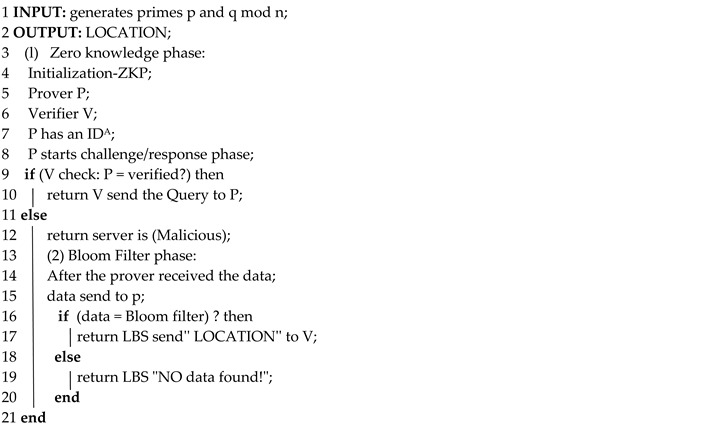


Step 2: Prover has an ID^A^ that can calculate J(A) = *f*(Id(A)) mod *n*. It also has a secret ID and the secret data (A) = J(A)^−*s*^.Step 3: Prover generates primes p and q such that p ≡ 3 mod 4 and q ≡ 3 mod 4. Prover computes n = pq and sends n to Victor.Step 4: The actual protocol BL0K begins from this step. Prover sends its identity ID J(A) after Prover selects a random secret *r* and *x* = *r^v^* mod *n* to Verifier.Step 5: Verifier sends a random challenge to Prover after Verifier selects a random challenge *e* in {1, 2, ..., *v*}.Step 6: Prover can compute then sends the following response to Verifier: *y*= *r* · secret(A)*^e^* mod *n*.Step 7: Verifier receives *y*, constructs J(A) = *f*(Id(A)) mod *n*, computes *z* = J(A)*^e^y^v^*, and accepts this round if *z* = *x* mod *n*.Step 8: The (GQ) zero-knowledge protocol has complete its work.Step 9: Verifier checks if the Prover is verified or not. If verified, the query send data query to Prover to obtain the location if data is present in bloom filter; otherwise, send a message “data not found”!Step 10: The BL0K protocol is completed.

Note, the rest of the code is used for result collection. Figure 7 shows the BLOT sequences methods for querying friends’ locations within LBS. Figure 8 shows the improvement with the BL0K approach.

## 5. BL0K Performance Measures

The three measures of performance indicate that Query1, Query 2, and Query 3 can be used as system security measures. A functional performance measure has already been introduced, referred to as performance entropy (PE). Therefore, both procedures have packets that tend to run amid the LBS server and the user and vice versa.

Entropy is often associated with L, which is the length of the messages [28,63]. The packets content can be split between the message/carrying data as well the non/of the message carrying data. The entire length of the whole of packages in a single exchange is L, which can be written as:(7)L = C + N
where C indicates the total of the lengths of the message/carrying subsets and N is the total of the lengths of the subset that does not carry any messages. The protocol’s performance entropy is:(8)$$PE\;=\frac{N}{L}$$

The highest PE value will increase the complexity of the entropy. For a given scheme, this measurement should be as small as possible. For instance, if all the information is conveyed in the clear, then PE = 0 and N = 0 and there is no entropy, which can be an insecure situation [63]. See Appendix A for the details.

### Attack Model

Of the four entities involved, CT is considered the most trusted entity. The cellular tower for each mobile phone is generally aware of the owner’s name and location in real-time; therefore, no attempt is made to conceal the devices’ locations from the networks. The security risks from LBS, SNS, and mobile users are of great concern.

BL0K assumes that either LBS or SNS might become compromised and seized by an adversary seeking to link users’ identities with their locations. However, the adversary cannot take control of both the LBS and SNS. Controlling the servers simultaneously is impossible, since they cannot collude with each other, preventing the adversary from controlling both servers simultaneously. In this case, the LBS server broadcasts the coverage region and the verifier connects. Then, the challenge/response cycle continues until a failed response is detected, or until the verifier chooses to discontinue. Note that the verifier needs to actively make a choice (by a button press, for example); the verifier may decide that it is sufficiently close to target (as given by a global positioning system (GPS), for example) so no further iterations are needed. The verifier may also coordinate responses given by more than one LBS server.

The BL0K supposes that LBS and SNS are both honest but still curious, and scans them to retrieve additional privacy information. Location privacy is threatened by the LBS that is continuously searching for the user’s location. In the BL0K framework, the private location information is protected from the LBS through the security property of the BF within the friends’ locations query. The SNS does not directly release the friend list to LBS. Instead, the SNS builds a BF and then deposits each element from the friend list into BF, which is later sent to the LBS. Once received, the LBS can then discover the potential friend list shared by the SNS through the testing of every element, which satisfies some condition for membership in BF. Since the determined input is impractical to enumerate, the users cannot maliciously exploit the friend list, preventing a violation in location privacy. However, the BF can query the elements and determine which one belongs to their respective friend list. At the conclusion of the BF process, the individual elements are no longer preserved. This mechanism structure is equipped with sound security measures. The BF is composed of a simple structure that can be established quickly. This feature of the filter is due to the insert and test data both using a hash function test. BL0K requires less communication and computation overhead than previous methods [21,27].

Finally, the private location information is hidden from the LBS by the BL0K framework. Thus, we proved that the private location information is protected and secured by BL0K.

## 6. Calculate the Performance Entropy for BL0K

We used the following table for BL0K performance measurement. The information sizes in bits) are outlined in Table 3.
$$H_{BLOT}\left(Avg\right):\;simulation\;founds\;to\;be=0.9756$$
while:$$H_{BL0K}\left(Avg\right):\;simulation\;founds\;to\;be=0.99$$

The PE Average of BL0K is 0.99, which means that the BL0K is better than both BLOT and BMobishare because a BLOT queries multiple data, of which only one message is relevant and can be decrypted correctly, as shown in Figure 9.

## 7. Performance Results

The conclusion is grounded in the notion that the assembly and presentation of computing have the following descriptions. Connection performance is defined as the time required by the prover to create a link to the verifier to begin a more protective exchange of information using any given procedure.

### 7.1. Connection Performance

Here, BLOT is equal to the BL0K, and BL0K/BLOT does not need to establish any connection before querying a location of a friend, hence the time of connection is 0 s.

### 7.2. Computing Performance

Computing performance can be viewed as the time a PC takes to launch an algorithm. Here, performance was calculated grounded on the packets exchanged in every protocol. This is due to the fact that the larger the number of packets involved, the higher the number of calculations needed to process them. However, in BLOT, the number of messages (i.e., seven messages) are needed to communicate the location of a friend. Additionally, in BL0K, seven messages are transmitted to query user location. Therefore, BLOT’s connection performance is equal while computing performance is comparable, as shown in Table 4.

### 7.3. Simulation Result for Computing Performance

Figure 10 shows the information necessary for the BL0K model, which is the real world scenario of the proposed model simulated in the OMNET++ environment. In this simulation, C++ code from eclipse has been used in the OMNET++ environment, and the network scenario consists of USER, CT, SNS, and LBS.

A dataset was used for the bloom filter and query-location information from SNS/LBS was used as a dummy dataset. We used the list of about 500 words of different sizes and then used those words to fill the bloom filter. A subset of the same list was used as data to query from SNS/LBS that was used in our maps or location-based application.

Figure 11 shows that the computation performance of BL0K is the best because it is fastest in the execution of one complete query. That occurred because the data flow in BL0K is less than in BLOT and BMobishare.

### 7.4. Computing Performance Measurement

Regarding computing performance measurement, the average of the BLOT computation time measured in the simulation process was about 10 ms. The BL0K shows an average calculating time of 7.8 ms. Both of BL0K and BLOT were measured at 2.5 GHz with a RAM of 4 GB.

## 8. Conclusions and Future Work

This paper outlined increasingly comprehensive research on the preservation of privacy in LBS. We proposed a BLoom Filter 0 Knowledge (BL0K) framework that aims to preserve the user’s privacy queries in LBS, consisting of both a bloom filter and zero-knowledge proof. We showed that BL0K enhances security and preserves privacy. BLOT is another mechanism, valuable in its ability to address the problems in LBS. However, this particular mechanism is not perfect, and may contain the potential risk of leakage. The BL0K approach is able to detect malicious servers directly, and can reduce the amount of data being exposed during an attack, thus slowing data leakage from the BF from 0.5 bit per query to 0.5 bit per N queries.

In a worst case scenario, an unauthorized LBS provider, through multiple attacks, is able to obtain a user’s complete location information history. A potential solution to this risk is to use the BL0K capability to mask any sensitive data exchanges occurring during communication of location-sharing procedures. We showed that the performance of the BL0K framework, using a dataset, is very close to the real-world queries by comparing its performance with the BLOT in terms of privacy (entropy). As shown in Figure 11, the computation performance of BL0K is the best because it requires the least time in the execution of one complete query because the data flow in BL0K is less than in BLOT and BMobishare.

The proposed method resulted in performance improvements up to 93% (average 25% to 30%). As a result, an untrusted participant cannot gain access to unauthorized privacy information. It follows that, when compared to BMobishare and BLOT, BL0K is the securest and most efficient service with the best performance.

For future work, we propose a cloaking system model called zero-knowledge cloaking (ZKC). In this model, it is possible for a malicious LBS to learn at most one bit of information about the location of the user based on a single query but can learn at most n bits about the location of the user at a past time after n consecutive queries. The ZKC system uses zero-knowledge proofs of location (ZKPL), so that the user can detect malicious LBS providers with only a few queries. This system can provide provably better anonymization than k-anonymity, and does not use non-person proxies (“dummies”).

## Figures and Tables

**Figure 1 sensors-19-00696-f001:**
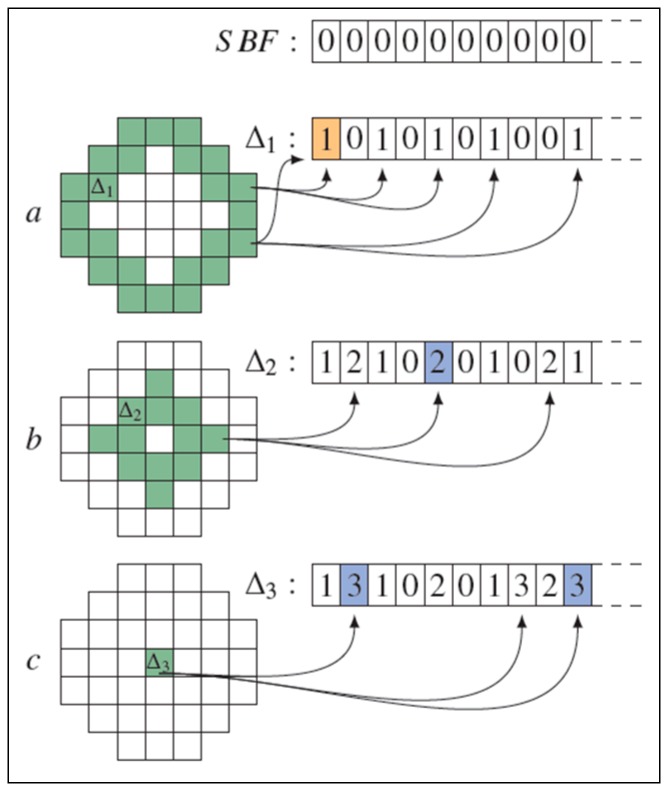
Spatial bloom filter (SBF). (**a**) Two BF elements joined to ∆_1_ are processed through the use of the hash functions, leading to six one-value elements that can then be written into the SBF. The same method can be used in (**b**) and (**c**).

**Figure 2 sensors-19-00696-f002:**
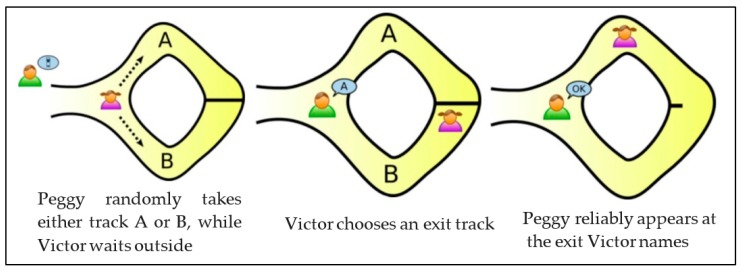
Zero Knowledge protocol example [50].

**Figure 3 sensors-19-00696-f003:**
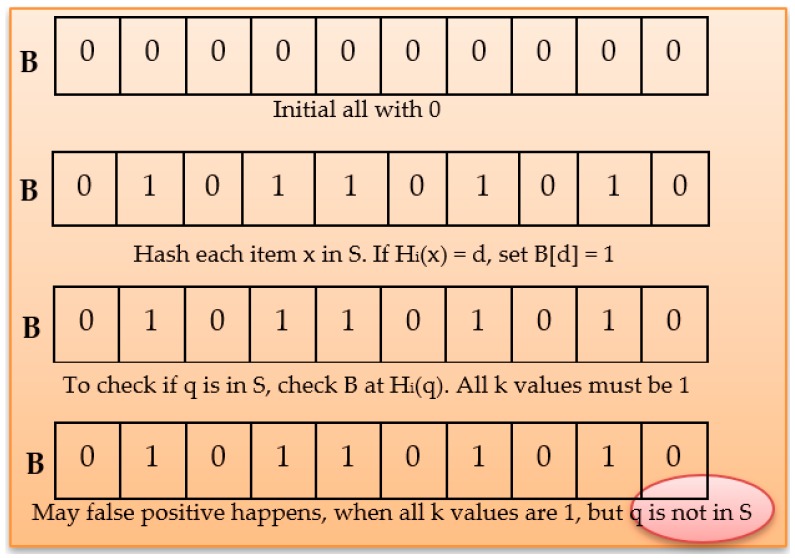
Standard Bloom Filters.

**Figure 4 sensors-19-00696-f004:**
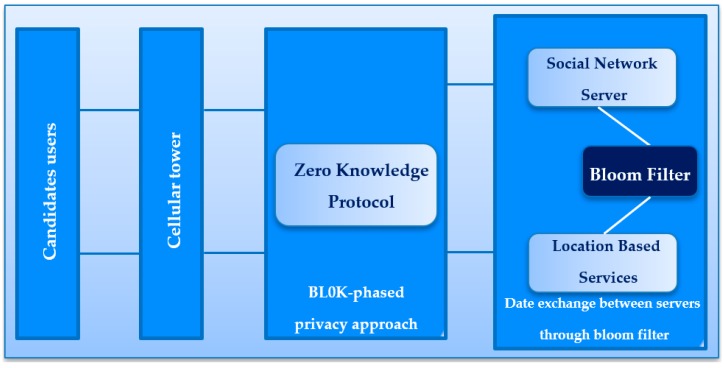
The overview of BL0K in LBS.

**Figure 5 sensors-19-00696-f005:**
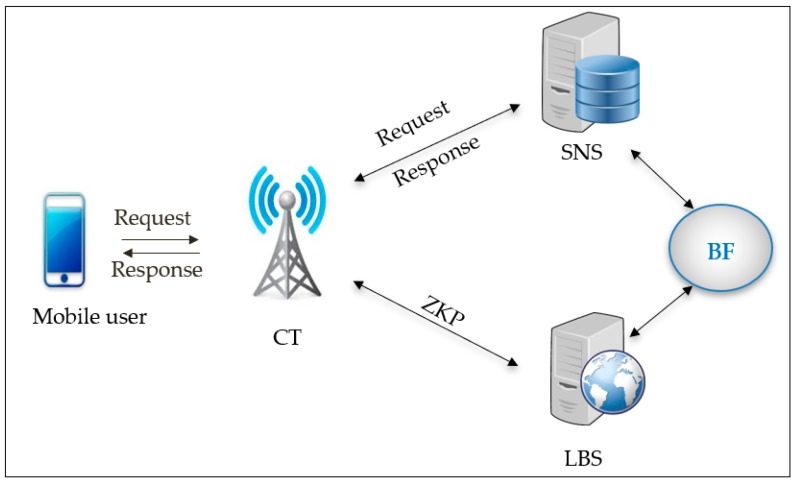
BL0K architecture in LBS.

**Figure 6 sensors-19-00696-f006:**
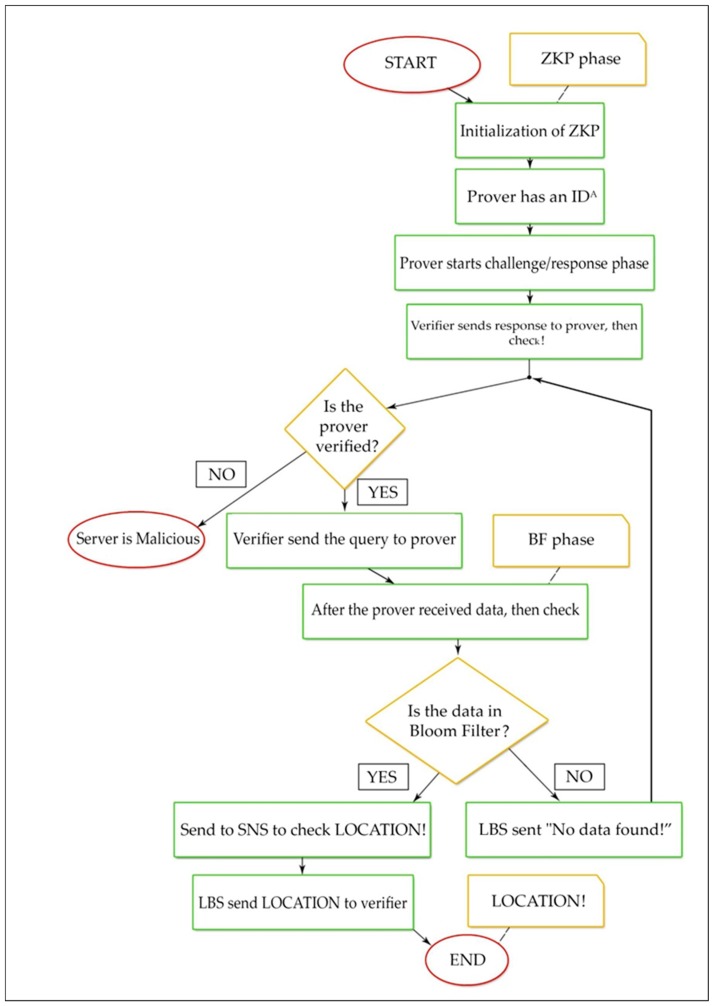
Flowchart of the BL0K approach.

**Figure 7 sensors-19-00696-f007:**
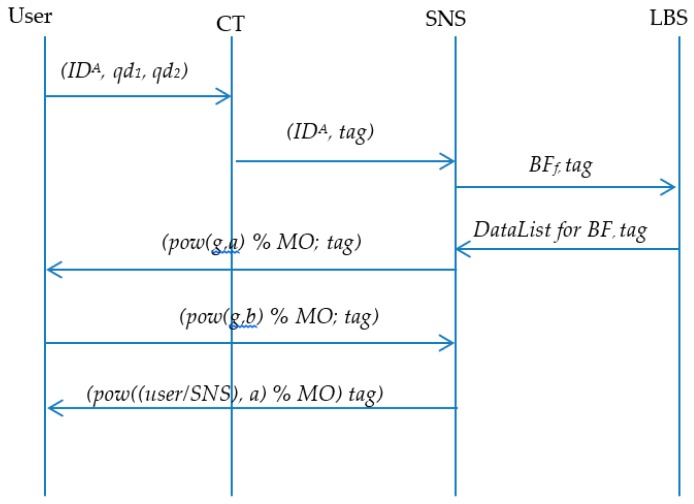
BLoom filter + Oblivious Transfer sequences of querying users locations.

**Figure 8 sensors-19-00696-f008:**
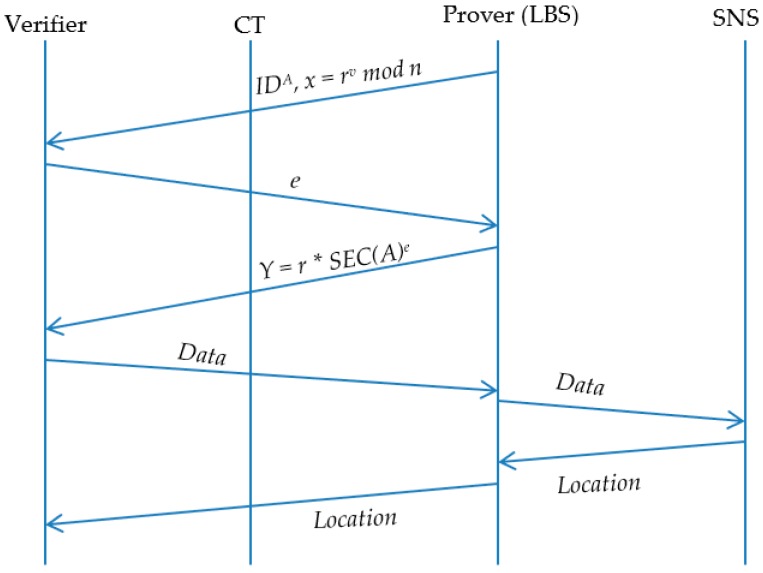
BL0K sequences of querying users’ locations.

**Figure 9 sensors-19-00696-f009:**
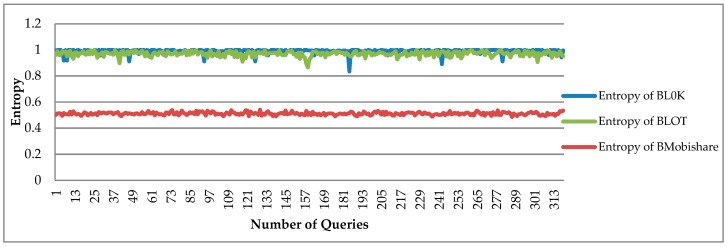
Performance entropy comparison of BL0K offers has better entropy than BLOT because the BLOT uses a lot of anonymous messages for increasing security, but in BL0K is achieved by using the zero-knowledge proof algorithm. No multiple or redundant information or queries sent on the network.

**Figure 10 sensors-19-00696-f010:**
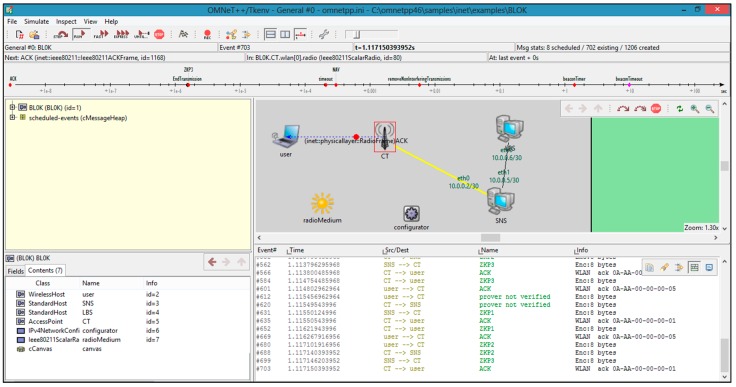
A snapshot of BL0K modeling.

**Figure 11 sensors-19-00696-f011:**
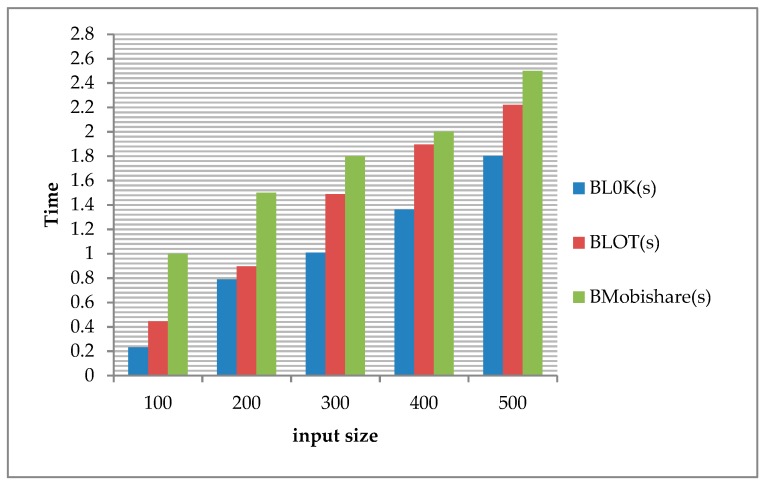
Computation performance comparisons.

**Table 1 sensors-19-00696-t001:** System architecture.

Symbol	Description
Verifier	Mobile device user
Prover	Location-based servers
SNS	The social network server
CT	Wi-Fi tower
BF	Bloom filter
ZKP	Zero Knowledge Proof

**Table 2 sensors-19-00696-t002:** Notations.

Symbol	Description
ID^A^	User A should be unique-identifier
PubMO^A^	The Public-modulus
Pube^A^	The Public-exponent
ID^CT^	Wi-Fi tower’s-identifier
Pub^Key^	Public key/coding-encoding
Skey^(AES)^	Symmetric key/coding-encoding

**Table 3 sensors-19-00696-t003:** Information size in bits.

Information	Size of Information (Bits)
J(A)	64
SNS to the user (X)	64
User to SNS (e)	32
SNS to the user (Y)	64
User to LBS and back (Location)	32

**Table 4 sensors-19-00696-t004:** Computation performance measurement.

Protocol	Computation Time	Connection in Time (s)
BMobishare	1.75 s	1.5
BLOT	10 ms	0
BL0K	7.8 ms	0

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
