# Peer review of "BL0K: A New Stage of Privacy-Preserving Scope for Location-Based Services"

_sensors, 2019, doi:10.3390/s19030696_

Reviewer 1 Report

The problem studied - preserving privacy in location -based services, is of great importance, and the authors may have proposed a good attempt to address the issue.  But the presentation of the paper, e.g., the words used, makes it hard to follow the paper and understand what others have done.  The following lists a few examples of the issues,

1. Line 46 - "these queries ... lead to issues with security" - what "security" issues are referred here?  Since "security" is of a very broad concept, it is hard for someone who is not an expert in the field to understand the "security" issues.

2. Line 73 - "servers can be penetrated resulting in mischievous activities.  Servers can generate queries to responses that cannot be altered."  Is the second sentence an example of the "mischievous activities"?  And what are these "queries" referring to?  As I thought servers only response to queries?

3. Line 109 - what exactly is an "problematic area"?

4. Line 136 - what are the "stated method's restrictions"?

5. Lines 148 - 150 - "set of connection protocols to stop the Mobishare+" and "Mobishare+ should be able top fulfill many multi[art coding and decoding" - is Mobishare+ a protocol for resolving the security issues or the other set of connection protocols doing the job?

6. When the authors mention "servers", do they mean server for providing LBS services or something else?  It is quite confusing when the problem is not clearly defined in the introduction part while it mentions any security issues there.

7. Line 219 - "... are an of the very active research area."

8. Some terms are also used without defining first, like "N" in line 117, "SBF" in Line 171, "FHE" in Line 220

9. Lines 285 - 292 - it is really confusing after reading "the prover P must be honest" .. "V can conduct a DoS attack ... P cannot determine any component of the k-bit vector K possessed by P" .. "... malicious prover is incapable of forging information possessed by the prover P".  Does it mean the prover can be malicious also?

10. Lines 295 - 297 and Lines 348 - 350 - "In the LBS scenario, the user plays the role of the prover P ..." and "the ZKP in the LBS scenario grants the user the role of the verifier V" - is the user the prover or the verifier in ZKP in the LBS scenario?

If the authors can present their work more clearly, it would be better for readers to follow and appreciate their work and contributions.

Author Response

Cover letter for

Manuscript ID: sensors-393872

 Please check the new manuscript with revisions are below:

1.    Line 20-22,  modified, to show the analysis clearer within the abstract.

2.    Line 46-57, I have added the security issues overlap with privacy. To improve the introduction.

3.    Line 83-87, modified the cases of servers when it is under full control by the attacker. To improve the introduction.

4     Line 88-95, modified the research [21] to improve the introduction.

5.    Line 101- 104, modified the research [22] to improve the introduction.

6.    Line 120-141, modified the contributions to improve it.

7.    Line 153-155, modified the word "restriction' to "strictness" to correct the sentence.

8.    Line 166-170,  updated the Mobishare+ [27].

9.    Line 190, I added the definition of SPF.

10.  Line 209-213, updated the research [35] to be more clear.

11.  Line 230, I added the definition of LSP.

12.  Line 239, corrected the sentence.

13.  Line 240, I added the definition of FHE.

14.  Line 279-282, updated.

15.  Line 295, I corrected the reference.

16.  Line 305-315, The prover has two cases, and I modified to be more clear.

17.  Line 317, I modified that the prover is LBS and the verifier is a user.

18.  Line 342-346, updated the Figure description.

19.  Line 361-371 and 380-387, Modified, to improve the research design.

20.  Line 98-99, Modified, to improve the research design.

21.  Line 509, corrected the Figure 6 name.

22.  Line 513, Figure 7 design updated.

23.  Line 531-556, modified the scenario to improve the methodology.

24.  Line 579, Table 3 modified to improve the methodology.

25.  Line 587, Figure 10 design updated.

26.  Line 612, corrected the Table 4 name.

27.  Line 615-635, Modified to improve the conclusion to be supported by results.

Reviewer 1:

Open Review

English language and style

( ) Extensive editing of English language and style required 
(x) Moderate English changes required 
( ) English language and style are fine/minor spell check required 
( ) I don't feel qualified to judge about the English language and style 

Yes

Can be improved

Must be improved

Not applicable

Does the introduction provide   sufficient background and include all relevant references?

( )

( )

(x)

( )

Is the research design   appropriate?

( )

(x)

( )

( )

Are the methods adequately   described?

( )

(x)

( )

( )

Are the results clearly presented?

( )

(x)

( )

( )

Are the conclusions supported by   the results?

( )

(x)

( )

( )

Comments and Suggestions for Authors

The problem studied - preserving privacy in location -based services, is of great importance, and the authors may have proposed a good attempt to address the issue.  But the presentation of the paper, e.g., the words used, makes it hard to follow the paper and understand what others have done.  The following lists a few examples of the issues,

1.      Line 46 - "these queries ... lead to issues with security" - what "security" issues are referred here?  Since "security" is of a very broad concept, it is hard for someone who is not an expert in the field to understand the "security" issues.

MODIFIED

Response 1: The first issue with implementations: whether data-stores (services, databases, and Web sites) can be queried in order to display a minimal amount of information, to the data-store while providing maximum utility (availability), to the user making the queries.

If you assume that the server may be compromised or even malicious, a second issue arises: is it possible to make a query with unforgeable response, which means that the server is unable to generate a fake response that seems to be a legitimate response. If the server generates a fake response, the recipient of the response may recognize that it is fake (integrity).

The third issue is that any information leak in the security system is not good. If each query leaks a bit and thousands of queries are made, the attacker can learn some information that must remain confidential (confidentiality). The Bloom filter, which is discussed in this paper is an example [6-9].

.

2.      Line 73 - "servers can be penetrated resulting in mischievous activities.  Servers can generate queries to responses that cannot be altered."  Is the second sentence an example of the "mischievous activities"?  And what are these "queries" referring to?  As I thought servers only response to queries?

MODIFIED

Response 2: One alarming issue is that the servers can be penetrated hence the servers will be under full control by the attacker, then resulting that to mischievous activities, including generating queries to responses and respond to queries. In case that server can generate queries to responses that cannot be altered. However, if any alteration takes place, the change, which is a fake response, can be identified by the recipient [18-20].

3. Line 109 - what exactly is an "problematic area"?

Response 3: The servers when become infected with malicious software.

Proposing a framework namely BLoom Filter 0 Knowledge (BL0K) includes the creation of an entirely new encryption communication procedure in a problem domain, which is with servers when become infected with malicious software, and the development of a secure application that can be used in the resource-constrained schemes such as mobile devices.

MODIFIED

4.      Line 136 - what are the "stated method's restrictions"?

Thanks for your valuable comments, It's strictness not" restrictions" and I have changed it.

Response 4: Though the prior work resolved the lack of flexibility in the sharing of the locations regarding privacy protection, the stated method's strictness stops when it has direct use [24].

5.      Lines 148 - 150 - "set of connection protocols to stop the Mobishare+" and "Mobishare+ should be able to fulfill many multi [art coding and decoding" - is Mobishare+ a protocol for resolving the security issues or the other set of connection protocols doing the job?

MODIFIED

Response 5: Li et al. came up with the idea of using Mobishare+ mechanism that tends to employ a remote set of connection protocols to stop the social network server from learning individual data. Though this mechanism has been improved the security, Mobishare+ needs to execute many complex coding and decoding operations which increase the computations overhead [27].

6.      When the authors mention "servers", do they mean server for providing LBS services or something else?  It is quite confusing when the problem is not clearly defined in the introduction part while it mentions any security issues there.

MODIFIED

Response 6: Yes, the servers here for providing LBS services. The domain of privacy partially overlaps with security (confidentiality, availability and integrity), which can include the concepts of appropriate use, as well as protection of information.

I have modified the problem, and I hope that to be more clear in the introduction. Thanks.

7. Line 219 - "... are an of the very active research area."

MODIFIED

Response 7: Privacy-preserving data exchange algorithms are very active research areas.

8. Some terms are also used without defining first, like "N" in line 117, "SBF" in Line 171, "FHE" in Line 220

MODIFIED

Response 8: N number of queries

SBF Spatial Bloom filter

FHE Fully homomorphic encryption

9. Lines 285 - 292 - it is really confusing after reading "the prover P must be honest" .. "V can conduct a DoS attack ... P cannot determine any component of the k-bit vector K possessed by P" .. "... the malicious prover is incapable of forging information possessed by the prover P".  Does it mean the prover can be malicious also?

MODIFIED

Response 9: The prover here, will take two of cases: one to be the honest and second case to be malicious.

Note that while the prover P must be honest in the first case, it is not necessary for the verifier V to be honest. V can conduct a denial-of-service attack against the prover P by providing inaccurate reports on the outcome of a particular ZKP iteration, but P cannot determine any component of the k-bit vector K possessed by P with better chances randomly.

Furthermore with second case, even if a malicious prover P offers a putative vector K', a single iteration of the ZKP algorithm can be used to demonstrate that K' == K with random probability at best (here denotes the I-th bit of the vector). Thus, the malicious prover is incapable of forging information possessed by the prover P and is incapable of forging proof of knowledge of that information either. After N iterations of the ZKP algorithm, the malicious prover has only a probability of 2-Nk of providing knowledge or proof of knowledge for a k-bit vector known as the prover P [54].

10. Lines 295 - 297 and Lines 348 - 350 - "In the LBS scenario, the user plays the role of the prover P ..." and "the ZKP in the LBS scenario grants the user the role of the verifier V" - is the user the prover or the verifier in ZKP in the LBS scenario?

MODIFIED

Response 10: User =Verifier

Prover = LBS

The ZKP approach has immediate relevance to the LBS privacy problem. In the LBS scenario, the user plays the role of the verifier V, while the LBS server plays the role of the prover P.

Again really thanks for your valuable comments and suggestions to enhance my paper. 

Reviewer 2 Report

Not very clear what is the significant contribution. Entropy enhancement over BLOT is only marginal.

Author Response

Cover letter for

Manuscript ID: sensors-393872

 Please check the new manuscript with revisions are below:

1.    Line 20-22,  modified, to show the analysis clearer within the abstract.

2.    Line 46-57, I have added the security issues overlap with privacy. To improve the introduction.

3.    Line 83-87, modified the cases of servers when it is under full control by the attacker. To improve the introduction.

4.    Line 88-95, modified the research [21] to improve the introduction.

5.    Line 101- 104, modified the research [22] to improve the introduction.

6.    Line 120-141, modified the contributions to improve it.

7.    Line 153-155, modified the word "restriction' to "strictness" to correct the sentence.

8.    Line 166-170,  updated the Mobishare+ [27].

9.    Line 190, I added the definition of SPF.

10.  Line 209-213, updated the research [35] to be more clear.

11.  Line 230, I added the definition of LSP.

12.  Line 239, corrected the sentence.

13.  Line 240, I added the definition of FHE.

14.  Line 279-282, updated.

15.  Line 295, I corrected the reference.

16.  Line 305-315, The prover has two cases, and I modified to be more clear.

17.  Line 317, I modified that the prover is LBS and the verifier is a user.

18.  Line 342-346, updated the Figure description.

19.  Line 361-371 and 380-387, Modified, to improve the research design.

20.  Line 98-99, Modified, to improve the research design.

21.  Line 509, corrected the Figure 6 name.

22.  Line 513, Figure 7 design updated.

23.  Line 531-556, modified the scenario to improve the methodology.

24.  Line 579, Table 3 modified to improve the methodology.

25.  Line 587, Figure 10 design updated.

26.  Line 612, corrected the Table 4 name.

27.  Line 615-635, Modified to improve the conclusion to be supported by results.

Reviewer 2:

Open Review

English language and style

( ) Extensive editing of English language and style required 
( ) Moderate English changes required 
(x) English language and style are fine/minor spell check required 
( ) I don't feel qualified to judge about the English language and style 

Yes

Can be improved

Must be improved

Not applicable

Does the introduction provide sufficient background   and include all relevant references?

( )

(x)

( )

( )

Is the research design appropriate?

( )

( )

(x)

( )

Are the methods adequately described?

( )

( )

(x)

( )

Are the results clearly presented?

( )

(x)

( )

( )

Are the conclusions supported by the results?

( )

( )

(x)

( )

Comments and Suggestions for Authors

1.      Not very clear what is the significant contribution. Entropy enhancement over BLOT is only marginal.

Response 1: The entropy just a privacy metric, to get better performance than other approaches, thus increases the privacy between users and servers. Therefore, we used BL0K to make the outputs of the algorithm appear to be completely random. This makes it computationally intractable to extract the plaintext from the queries, while at the same time still allowing the algorithm to work correctly in a probabilistic sense (where the probability of error can be made arbitrarily small). Also we used the network traffic to measure the network bandwidth and found that the BL0K consume less network bandwidth than BLOT and BMobishare.

The traffic result is below, if you advise I can add it in the paper.

Thanks for your valuable comments, so I have modified the contributions to be more clear:

 Proposing a framework namely BLoom Filter 0 Knowledge (BL0K) includes the creation of an entirely new encryption communication procedure in a problem domain, which is with servers when become infected with malicious software, and the development of a secure application that can be used in the resource-constrained schemes such as mobile devices.

 The BL0K initiates the response phase. Usually, this phase is executed multiple times. Therefore, any failure of the response matching to the challenge would result in a malicious server. Additionally, it will protect the authentication between the server and the users of the mobile devices against attacks (malicious LBS). The idea is that the BL0K will have the ability to minimizing the amount of information an attacker can obtain.

Therefore, the approach used in BL0K can reduce the amount of data that is being exposed during an attack, thus slowing data leakage from the BF from 0.5 bit per query to 0.5 bit per N queries. On average, where N is arbitrarily large. Therefore, BL0K can be used to lessen the data leakage.

We use BL0K to make the outputs of the algorithm appear to be completely random. This makes it computationally intractable to extract the plaintext from the queries, while at the same time still allowing the algorithm to work correctly in a probabilistic sense (where the probability of error can be made arbitrarily small) and this increases the level of performance in general, and this would raise the level of privacy to a high level.

Evaluating the performance of BL0K framework:

o  Using  a dataset close to the real-world queries by comparing its performance with the BLOT  in terms of privacy (entropy).

o  The reported performance improvements are gone up to 93% (average 25% to 30%).

Explanations: As can be seen from the graph that BL0K consume less network bandwidth than BMobishare and BLOT. BMobishare uses a lot of anonymous messages for increasing security, but in BLOT is achieved by using encryption algorithms AEC256. While BLOK is being achieved by using zero knowledge proof algorithm, no multiple or redundant information or queries is sent on to the network. So, the reported network traffic improvements are gone up to 100% within some of the queries (average 35% to 40% of the queries).

Network traffic improvement percentage =

(sum of BLOT traffic- sum of BL0K traffic)/ sum of BLOT traffic*100

=( 76372 - 47634)/ 76372*100

=38%

Please find the network traffic (Figure) in attached. Thanks

Reviewer 3 Report

English language changes are required in several parts of the paper. Images (quality, font enlarged) must be improved. Comparison (in terms of performance) with other approaches would help.

some specific comments below.

Page 2, line 120:

"Proposing a framework namely BLoom Filter 0 Knowledge (BL0K) includes the creation of 120 an entirely new encryption communication procedure in a problem domain, which is with servers 121 when become infected with malicious software, and the development of a secure application that 122 can be used in the resource-constrained schemes such as mobile devices"

The sentence must be re-phrased. It's too long. Shouldn't start with "Proposing"maybe "We propose .."

Also the next phrases after this one have to be reviwed. Why are they bulleted ? The authors should resume in brief their contribution not decribe the approach ..

I think the lines from 120 to 141 on page 3 have to be reviewed both for English and content.

Page 4, line 156: "had come up  with ". It has to be reviviwed. Better "came up with ..

Page 4, line 172 : "divulges" . Not used, replace with some synonym.

Page 5, line 209: "BF was proposed as a probabilistic data structures to hiding sensitive data"It must be re-phrased. Better "BF has been proposed ... to hide ..."

Figure 2. "ZKP example" on page 7. Font is too small, the figure must be at least enlarged it's not easy to see the details.

Figure 6 on page 13. Font is too small, it's difficult to follow the flowchart.Also Figure 8 on page 14 has the font too small. Figure 9 on page 15 has all the font in bold. Why ?

Page 17, line 625: "In a worst-case scenario, imagines an unauthorized LBS provider, through multiple attacks, is 625 able to obtain a user’s complete social relations history". It must be re-phrased. English language errors.

Page 17, line 617: "Additionally, this paper proposing a framework namely BLoom Filter 0 Knowledge (BL0K), 617 which aims to preserve the user's privacy queries for in LBS, based on Bloom Filter and a Zero-618 Knowledge Proof." It must re-phrased. English language errors.

Page 16. I think more detials have to be given on how the performance measurements have been done. I means it seems that the authors evaluated the performance based on the number of messages exchanged, It's not enough. Have they created a simultation test bed in a controlled environment or how did they measure the performance of the other protocols ? If it's based only on the number and size if the messages maybe it's not enough. I think this section should be expanded with further details, it doesn't convince me.

Author Response

Reviewer 3

Open Review

English language and style

( ) Extensive editing of English language and style required 
(x) Moderate English changes required 
( ) English language and style are fine/minor spell check required 
( ) I don't feel qualified to judge about the English language and style 

Yes

Can be improved

Must be improved

Not applicable

Does the introduction provide   sufficient background and include all relevant references?

( )

(x)

( )

( )

Is the research design   appropriate?

( )

(x)

( )

( )

Are the methods adequately   described?

( )

(x)

( )

( )

Are the results clearly   presented?

( )

(x)

( )

( )

Are the conclusions supported by   the results?

( )

(x)

( )

( )

Comments and Suggestions for Authors

English language changes are required in several parts of the paper. Images (quality, font enlarged) must be improved. Comparison (in terms of performance) with other approaches would help.

MODIFIED

some specific comments below.

Page 2, line 120:

"Proposing a framework namely BLoom Filter 0 Knowledge (BL0K) includes the creation of 120 an entirely new encryption communication procedure in a problem domain, which is with servers 121 when become infected with malicious software, and the development of a secure application that 122 can be used in the resource-constrained schemes such as mobile devices"

The sentence must be re-phrased. It's too long. Shouldn't start with "Proposing"maybe "We propose .."

Response 1: We propose a framework called (BL0K) that creates an entirely new encryption communication procedure in the problem domains (malicious server) and develops a secure application for used in resource-constrained schemes such as mobile devices.

Also the next phrases after this one have to be reviwed. Why are they bulleted ? The authors should resume in brief their contribution not decribe the approach ..

I think the lines from 120 to 141 on page 3 have to be reviewed both for English and content.

Response 2:

·       We propose a framework called (BL0K) that creates an entirely new encryption communication procedure in the problem domains (malicious server), and develops a secure application for used in resource-constrained schemes such as mobile devices.

·       BL0K is capable of reducing the amount of exposed data during an attack, thus slowing data leakage from the BF from 0.5 bit per query to 0.5 bit per N queries. On average, where N is arbitrarily large.

·       The BL0K can be used to make the outputs of the algorithm to appear completely randomize. This feature makes it computationally intractable to extract the plaintext from the queries, while simultaneously allowing the algorithm to work correctly in a probabilistic sense (where the value of the probability of error is made arbitrarily small) and thus increasing the general performance level which would also raise the level of privacy higher.

·       Evaluating the performance of BL0K framework:

o  Using a dataset close to the real-world queries by comparinge its performance with the BLOT  in terms of privacy (entropy).

o  The reported performance improvements increased to 93% (averaging 25% to 30%).

Page 4, line 156: "had come up  with ". It has to be reviviwed. Better "came up with ..

Response 3: Modified.

Page 4, line 172 : "divulges" . Not used, replace with some synonym.

Response 4: Modified to disclose.

Page 5, line 209: "BF was proposed as a probabilistic data structures to hiding sensitive data"It must be re-phrased. Better "BF has been proposed ... to hide ..."

Response 5: BF has been proposed as a probabilistic data structure to hide sensitive data. Unfortunately, each request leaks a maximum of one bit of information and the hash function requires careful designing and security. The BF analysis is focused on the orthogonality and independence of the structure. The results show that the proposed hash functions are less dependent and permeable than the comparison method while noticing significant improvements in performance [35].

Figure 2. "ZKP example" on page 7. The font is too small, the figure must be at least enlarged it's not easy to see the details.

Response 6: Modified.

Figure 6 on page 13. Font is too small, it's difficult to follow the flowchart. Also Figure 8 on page 14 has the font too small. Figure 9 on page 15 has all the font in bold. Why ?

Response 7: Modified.

Page 17, line 625: "In a worst-case scenario, imagines an unauthorized LBS provider, through multiple attacks, is 625 able to obtain a user’s complete social relations history". It must be re-phrased — English language errors.

Response 8: In a worst-case scenario, an unauthorized LBS provider, through multiple attacks, is able to obtain a user’s complete locations information history.

Page 17, line 617: "Additionally, this paper proposing a framework namely BLoom Filter 0 Knowledge (BL0K), 617 which aims to preserve the user's privacy queries for in LBS, based on Bloom Filter and a Zero-618 Knowledge Proof." It must re-phrased — English language errors.

Response 9: In addition, this paper proposes a BLoom Filter 0 Knowledge (BL0K) framework, which aims to preserve the user's privacy queries in LBS, consisting of both Bloom Filter and Zero-Knowledge Proof.

Page 16. I think more detials have to be given on how the performance measurements have been done. I means it seems that the authors evaluated the performance based on the number of messages exchanged, It's not enough. Have they created a simultation test bed in a controlled environment or how did they measure the performance of the other protocols ? If it's based only on the number and size if the messages maybe it's not enough. I think this section should be expanded with further details, it doesn't convince me.

Response 10: Expanded has been done.

Thank you for your valuable comments and suggestions to enhance my paper. I appreciated that.

Round  2

Reviewer 1 Report

The authors have tried addressing my concerns raised previously,  Most of the concerns are appropriately addressed, but a few issues remain as follow.

Line 401 mentions LBS provides real-time locations of nearby people (which is feasible) but Line 555 mentions discovering friends' locations within LBS (which does not seem feasible as LBS only stores anonymized data (Line 400))?

The authors mention malicious servers (in general, like generating queries and respond to queries), but it is unclear to me in the LBS scenario, how a malicious server would behave. (I suppose it will not generate queries?)  And what if the malicious server is only curious about the users' locations, then it cannot be identified using ZKP?  Since it seems to me that the ZKP is only for verification but has nothing related to BF (i.e. locations queries of the users).

It is better if the authors can give a clear attack model of the malicious server to aid the explanation of their proposed solution.

A "second server" is mentioned in Line 327, but I am not sure about its purpose.

The use of "tends to", "often" between Lines 398 and 402 makes the roles (and responsibilities) of the SNS, LBS, and CT unclear to me.  For example, "CT tends to aid", does it mean CT will not aid at some instances?  Then what are these instances?

Author Response

Reviewer 1

Open Review

English language and style

( ) Extensive editing of English language and style required 
(x) Moderate English changes required 
( ) English language and style are fine/minor spell check required 
( ) I don't feel qualified to judge about the English language and style 

Yes

Can be improved

Must be improved

Not applicable

Does the introduction provide   sufficient background and include all relevant references?

( )

(x)

( )

( )

Is the research design   appropriate?

(x)

( )

( )

( )

Are the methods adequately   described?

( )

(x)

( )

( )

Are the results clearly   presented?

(x)

( )

( )

( )

Are the conclusions supported by   the results?

(x)

( )

( )

( )

Comments and Suggestions for Authors

The authors have tried addressing my concerns raised previously,  Most of the concerns are appropriately addressed, but a few issues remain as follow.

1.                  Line 401 mentions LBS provides real-time locations of nearby people (which is feasible) but Line 555 mentions discovering friends' locations within LBS (which does not seem feasible as LBS only stores anonymized data (Line 400))?

Response 1: In this sequence, the LBS can discover the possible friend-list locations shared with SNS within testing each element which fulfills some of the conditions for membership in Bloom filter. Because the input sets are impractical to enumerate, the users cannot maliciously exploit the friend-list to violate location privacy.

Maybe we can change the “discovering” word to “querying” to be more clear.

2.                  The authors mention malicious servers (in general, like generating queries and respond to queries), but it is unclear to me in the LBS scenario, how a malicious server would behave. (I suppose it will not generate queries?)  And what if the malicious server is only curious about the users' locations, then it cannot be identified using ZKP?  Since it seems to me that the ZKP is only for verification but has nothing related to BF (i.e. locations queries of the users).

It is better if the authors can give a clear attack model of the malicious server to aid the explanation of their proposed solution.

Response 2:

Attack model

Out of the four entities involved, CT is considered the most trusted entity. The cellular tower for each mobile phone is generally aware of the owner’s name and location in real-time, therefore, no attempt is made to conceal the devices' locations from the networks. The security risks from LBS, SNS, and mobile users are of great concerns.

The BL0K assumed that either LBS or SNS might become compromised and seized by an adversary, seeking to link users' identities with their locations. However, the adversary cannot take control of both the LBS and SNS. Controlling the servers simultaneously is impossible since they cannot collude with each other, making it impossible for the adversary to control both servers at the same time. In this case, the LBS server broadcasts coverage region and verifier connects. Then the challenge/response cycle continues until a failed response is detected, or until the verifier chooses to discontinue. Note that the verifier does need to actively make a choice (by a button press, for example); the verifier may decide that is sufficiently close to target (as given by GPS, for example) that no further iterations are needed. It may also happen that the verifier is coordinating responses given by more than one LBS server.

On the other hand, the BL0K is supposed that LBS and SNS are both honest, but still curious, and scan them to retrieve additional privacy information. Location privacy is threatened by LBS which is continuously searching for the user's location. In the BL0K framework, the location information of privacy is protected from the LBS through the security property of the BF within the friends' locations query. The SNS doesn’t release the friend list to LBS directly. Instead, the SNS builds a BF and then deposit each element from the friend-list into BF that is later sent to LBS. Once received, the LBS can then discover the potential friend-list shared by the SNS through the testing of every element, which satisfies some condition for membership in BF. Since the determined input is impractical to enumerate, the users cannot maliciously exploit the friend-list, preventing a violation in location privacy. However, the BF can query the elements and determine which one belongs to their respective friend-list. At the conclusion of the BF process, the individual elements are no longer preserved. This mechanism structure is equipped with sound security measures. Furthermore, the BF is composed of a simple structure that can be established quickly. This feature of the filter is due to the insert and test data both using a hash function test, BL0K incurs at lower communication and computation overhead than the previous method [27].

Finally, the location information of privacy is hidden from the LBS by the BL0K framework. Thus, we can prove that the location information of privacy is protected and secure by BL0K.

3.                  A "second server" is mentioned in Line 327, but I am not sure about its purpose.

MODIFIED

Response 3: It’s SNS server.

4.                  The use of "tends to", "often" between Lines 398 and 402 makes the roles (and responsibilities) of the SNS, LBS, and CT unclear to me.  For example, "CT tends to aid", does it mean CT will not aid at some instances?  Then what are these instances?

Response 4: Actually the CT is assumed as a trusted entity between the four entities: user, LBS and SNS. So that, I modified as follows:

·                     First, user A can gain entry with internet speed of either 3G/4G or 5G now, and later share her/his whereabouts and then ask about the friend's position.

·                     Secondly, SNS is able to manage the identity-related data of the user that include a backing store, friend lists, user profiles, etc.

·                     Thirdly, LBS often stores the anonymized data of the location of the user and gives the LBS, per the request of the user, the real-time locations of the nearby people.

·                     Lastly, the CT be oriented to aid the communication of the user with LBS and SNS.

Thank you for your valuable comments and suggestions to enhance my paper. I appreciated that.

Reviewer 3 Report

Section 7 has to be re-written in my opinion. It contains several errors, e.g. "In both BL0K and BLOT, there is a calculated time of 715 over 2.5 GHz and a RAM of 4GB." The calculated time is not expressed in GHz or in RAM. It's not clear how the measurement has been performed. The description of the simulation environment must be placed before the sections showing the results.

There are several English errors.

Page 1, line 20: "..performed ..".. "in performance": you need to reformulate to avoid repetitions.

Page 3, lines 121 - 135. Reformulate. The contributions have to be resumed in a concise and clear manner. As it is now it's a bulleted list mixing contributions and features.

Page 4, line 166: this paper disclose "..: English error

Page 7: Figure 2 is uncomprehensible. Where is P and where is V? 

Page 8, lines 326 - 338. Have to be reviewed. For example"PROBABILITY"is a feature of what? of even "SERVER RANGE". You refer to a second server but you used SNS term.

Figure 6 on page 13 contains English errors.

Author Response

Reviewer 3

Open Review

English language and style

(x) Extensive editing of English language and style required 
( ) Moderate English changes required 
( ) English language and style are fine/minor spell check required 
( ) I don't feel qualified to judge about the English language and style 

Yes

Can be improved

Must be improved

Not applicable

Does the introduction provide sufficient background and include   all relevant references?

( )

(x)

( )

( )

Is the research design appropriate?

( )

( )

(x)

( )

Are the methods adequately described?

( )

( )

(x)

( )

Are the results clearly presented?

( )

( )

(x)

( )

Are the conclusions supported by the results?

( )

( )

(x)

( )

Comments and Suggestions for Authors

1.    Section 7 has to be re-written in my opinion. It contains several errors, e.g. "In both BL0K and BLOT, there is a calculated time of 715 over 2.5 GHz and a RAM of 4GB." The calculated time is not expressed in GHz or in RAM. It's not clear how the measurement has been performed. The description of the simulation environment must be placed before the sections showing the results.

Modified

Response 1: Regarding computing performance measurement, the average of the BLOT computation time measured in the simulation process was about 10 ms. The BL0K shows an average calculating time of 7.8ms. Both of BL0K and BLOT were measured at 2.5 GHz with a RAM of 4 GB.

As reviewer’s advice, I have moved the description of the simulation environment before the results sections”

There are several English errors.

Regarding the English errors, I used MDPI editing services.

2.    Page 1, line 20: "..performed ..".. "in performance": you need to reformulate to avoid repetitions.

Modified

Response 2: Analysis of the results demonstrated that BL0K performance is decidedly better when compared to the referenced approaches using the privacy entropy metric.

3.    Page 3, lines 121 - 135. Reformulate. The contributions have to be resumed in a concise and clear manner. As it is now it's a bulleted list mixing contributions and features.

Modified

Response 3:

(1)   We propose a framework called BL0K that creates an entirely new encryption communication procedure in the problem domains (malicious server) and a secure application for use in resource-constrained schemes such as mobile devices.

(2)   BL0K is capable of reducing the amount of exposed data during an attack, hence BL0K can be used to lessen data leakage.

(3)   BL0K can be used to make the outputs of the algorithm appear completely randomized. Thus, increasing the general performance level that would also increase the level of privacy.

(4)   We evaluated the performance of BL0K framework: using a dataset close to the real-world queries by comparing its performance with the BLOT in terms of privacy (entropy), and the reported performance improvements increased to 93% (averaging 25% to 30%).

4.    Page 4, line 166: this paper disclose "..: English error

Modified

Response 4: Palmieri et al. [31] disclosed how the availability of cheap positioning systems has made it possible for them to be embedded in devices and other smartphones.

5.    Page 7: Figure 2 is uncomprehensible. Where is P and where is V? 

Response 5: Modified

6.    Page 8, lines 326 - 338. Have to be reviewed. For example "PROBABILITY" is a feature of what? of even "SERVER RANGE". You refer to a second server but you used SNS term.

Modified

Response 6:

The user, LBS server, and SNS should have the following properties for their protocols:

(1)   User/server: The user and SNS can send a particular query if there is something in the LBS server’s range.

(2)   Query list: The original server can locate one bit of data from the query list, while SNS can locate none.

(3)   Server range: Set of objects in the range of the server will need to be itemized.

(4)   Malicious LBS server: A malicious LBS server can be able to detect with probability q.

(5)   Probability: Probability q can be increased close to 1 by increasing the queries.

7.    Figure 6 on page 13 contains English errors.

Response 7: Modified

Thank you very much for your valuable comments to enhance my paper . I appreciated that.
